# Bovine Mineral Grafting Affects the Hydrophilicity of Dental Implant Surfaces: An In Vitro Study

**DOI:** 10.3390/ma17164052

**Published:** 2024-08-15

**Authors:** Allyson Newman, Nina K. Anderson, Georgios E. Romanos

**Affiliations:** 1Laboratory of Periodontal, Implant-, Phototherapy, Department of Periodontics and Endodontics, Stony Brook University, Stony Brook, NY 11794, USA; georgios.romanos@stonybrookmedicine.edu; 2Department of Oral Biology and Pathology, Stony Brook University, Stony Brook, NY 11794, USA; nina.anderson@stonybrookmedicine.edu

**Keywords:** dental implants, hydrophilicity, niobium, titanium, wettability, xenogenic bone graft, zirconia

## Abstract

Wettability is recognized as an important property of implant surfaces for ensuring improved biological responses. However, limited information exists on how bone grafting procedures including materials influence the hydrophilic behavior of implant surfaces. This in vitro study aimed to investigate the influence of two bovine grafting materials after hydration on the wettability of four different disk surfaces: commercially pure titanium (CP-Ti), titanium–zirconium dioxide (TiZrO_2_-Cerid^®^), zirconia (SDS^®^), and niobium. Wettability tests were performed on each of the four implant surfaces with a solution of 0.9% sodium chloride after mixture with W-bone^TM^ (Group A) or Bio-Oss^®^ (Group B) or 0.9% sodium chloride alone (Group C). In total, 360 contact angle measurements were completed with *n* = 30 per group. Statistical analysis was performed using a one-way analysis with variance (ANOVA) test with a significant mean difference at the 0.05 level. For pure titanium, Group A demonstrated increased hydrophilicity compared to Group B. Both TiZrO_2_ and zirconia showed significant differences for Groups A, B and C, exhibiting a decrease in hydrophilicity after the use of bovine grafting materials compared to titanium surfaces. Niobium remained consistently hydrophobic. In summary, this study revealed that bovine grafting materials may diminish the hydrophilicity of zirconia surfaces and exert varied effects on titanium and niobium. These findings contribute to the understanding of implant surface interactions with grafting materials, offering insights for optimizing biological responses in implantology.

## 1. Introduction

Since the first dental implant placed by professor Per-Ingvar Brånemark in 1965, titanium has remained the preferred material for dental implants due to its ability to establish an intimate direct interaction with bone, which is a phenomenon termed osseointegration [1]. Therefore, in dental implantology, commercially pure titanium and its alloys are the gold standard with excellent biocompatibility and other advantageous mechanical properties including corrosion resistance, high strength-to-weight ratio, high melting point, and small degree of thermal expansion [2]. However, despite the superior corrosion resistance of titanium, research has shown that after the insertion of titanium dental implants, deposits of fine titanium particles can migrate with the blood into tissues of internal organs, including the lungs and bones. This may trigger an inflammatory response that could play a contributing role in the development of peri-implantitis defects [3].

For this reason, metal-free implants have been explored as alternative options in dentistry. While the vast majority of dental implants today are still made from titanium, we know that other metals such as niobium may also obtain osseointegration and that some ceramic materials have shown adequate implant stability [4]. In recent years, zirconia has gained great clinical interest as a non-metal alternative implant material due to its esthetically superior white-colored surface [5], reduced potential to adhere to microorganisms [6], and excellent biocompatibility [3]. In fact, animal studies have shown that zirconia implants have evidenced-based osseointegration similar to that of titanium implants [7].

Additionally, surface coating technologies are an alternative method used to improve the osseointegrative properties of implant surfaces and can evade the issue of titanium ion release. In particular, a hybrid titanim–zirconia material consists of a titanium core encased in a layer of zirconium dioxic ceramic (Cerid^®^). This coating allows for increased corrosion resistance, enhanced biological compatibility, and the prevention of titanium ion release in order to decrease cytotoxicity [8]. The ceramic coating that covers the titanium also has the potential to protect against sensitivity to titanium in patients with allergic reactions to the metal.

When searching for novel implant surface materials, a physical property that researchers should take into consideration includes wettability, which describes the tendency of a liquid to spread across a solid implant surface. This surface property forms the basis for cell adhesion and is influenced by surface roughness. Wettability is inversely proportional to contact angle (*θ*), which can be measured experimentally between the tangent drawn at the three-phase solid/liquid/gas interface and the horizontal baseline of a substrate surface [9]. Contact angle measurements < 90° are considered hydrophilic and favorable, while surfaces with contact angles > 90° are regarded as hydrophobic and unfavorable [10]. Therefore, smaller contact angles indicate superior surface wettability. Generally, as wettability is increased, biocompatibility is enhanced, promoting interactions between dental implant surfaces and the biological environment. Some benefits of increased hydrophilicity include advantages during the initial stages of wound healing during osseointegration, improvements in bone–implant contact % (BIC%) and bone anchorage during healing, and increased initial cell attachment and spreading of osteoblastic cells on biomaterial surfaces [11,12].

Oftentimes, insufficient alveolar bone can negatively affect implant insertion and prognosis. Therefore, bone grafting materials are commonly utilized prior to or in conjunction with implant placement. However, limited information exists on the impact of dental bone grafting materials on the wettability of implant surfaces. For instance, Bio-Oss^®^ cancellous (Geistlich Biomaterials, Wolhusen, Switzerland) is the most well-known deproteinized cancellous bovine bone mineral (DBBM), which is composed of an osseous mineral obtained from spinal bone [13]. Bio-Oss^®^ has a porous structure similar to that of human bone. It has shown high biocompatibility with oral hard tissues in preclinical and clinical studies [14]. A significant advantage of xenografts is their slower resorption compared to autografts, which enables them to act as a scaffold to new bone formation during the regenerative process [15].

A relatively newer xenograft that has recently entered the US market is W-Bone^TM^ (Wishbone SA, Liége, Belgium), which has many similar properties to Bio-Oss^®^. Both materials are composed of deproteinized hydroxyapatite from bovine origin. They differ in their chemical composition, crystalline phases, dosage form, specific surface area, and packaging configuration. A main difference is that during the sintering process of W-Bone^TM^, magnesium oxide (MgO) naturally appears in the form of magnesium oxide crystals [16]. Little to no research exists on the comparison of how W-Bone^TM^ or Bio-Oss^®^ affect the wettability of dental implant surfaces. Variations in manufacturing procedures as well as physicochemical characteristics may lead to differences in hydrophilicity, which can subsequently impact the regenerative capabilities of these grafting materials. Therefore, the aim of this in vitro study was to test the null hypothesis that bovine grafting materials, after hydration, have no effect on the hydrophilicity of CP-Ti, TiZrO_2_, zirconia, and niobium implant disk surfaces compared to the saline control. 

## 2. Materials and Methods

This in vitro study was performed in the Laboratory of Periodontal, Implant, and Phototherapy at the Dept. of Periodontics and Endodontics at Stony Brook University School of Dental Medicine, Stony Brook, NY. Four different dental implant surfaces in the form of small circular disks were used:Commercially pure titanium grade II (Aaleberts Surface Technologies, Germany), 314.16 mm^2^ surface area;TiZrO_2_ (Cerid^®^) (Aaleberts Surface Technologies, Germany), 314.16 mm^2^ surface area;Zirconia (SDS^®^) (Swiss Dental Solutions, Germany), 113.1 mm^2^ surface area; andNiobium (Aaleberts Surface Technologies, Germany), 314.16 mm^2^ surface area. Disks coated with TiZrO_2_ or niobium were manufactured using cathodic arc deposition (arc-PVD) technology under process pressures ranging from 10^−2^ to 10^−3^ mbar. In this method, metal is evaporated from a target plate by an electric arc, while reactive gases such as oxygen or nitrogen are introduced. This results in the formation of positive ions of Ti, Z, Nb, N, and O, which are accelerated by an electric field. The coating is then deposited directly from the ions through a highly energetic process that surpasses measurable temperatures.

Assessment of the four surfaces was performed using three different wetting solutions:Group A: 0.9% sodium chloride after hydration with 0.25–1 mm W-bone^TM^ (Wishbone SA, Liege, Belgium);Group B: 0.9% sodium chloride after hydration with 0.25–1 mm small granule Bio-Oss^®^ cancellous (Geistlich Biomaterials, Wolhusen, Switzerland); andGroup C: 0.9% sodium chloride (control).

For experimental groups where a xenogenic bone graft was added, 0.5 g of each xenograft was mixed for 1–2 min and dissolved in a small quantity of 0.9% sodium chloride, as performed in clinical practice as standard of care. The hydration of each xenograft was performed via thorough mixing at room temperature until the particles adhered to one another, resulting in a residual amount of liquid containing dissolved grafting particles in the sterile saline solution. This remnant solution was aspirated and subsequently used for wettability testing.

Each disk was cleaned and dried at room temperature; then, it was subsequently positioned onto the mechanical stage of a contact angle goniometer (Ossila^®^, Sheffield, UK). Then, a calibrated microsyringe was used to dispense 7.5 microliters of the wetting solutions at the center of each disk. The drop placed on each surface was subsequently video-recorded. This resulted in 12 experimental groups, which were each composed of 15 repetitions (*N* = 360). Contact angle measurements were obtained on the left and right sides of each droplet. A prior power analysis was conducted using the approach of Cohen (1992) to determine the minimum sample size required to test the study hypothesis. Results indicated the required sample size to achieve 80% power for detecting a medium/large effect, at a significance criterion of α = 0.05, was *N* = 30 for ANOVA [17].

Wettability was determined via the sessile drop technique, which is the most common approach to gain insight into wetting behavior. This was performed using the Ossila^®^ contact angle goniometer (error ±1) and accompanying software. An inbuilt software-based fitting procedure was performed on images captured 1 s after droplet deposition on each sample surface to analyze CA values. For each trial, the Ossila^®^ v3.0.6.0 software traced the droplet edge and calculated the gradient of the tangent from the droplet edge to the point where it intersected the baseline. The contact angle between them was then calculated on both the left and right sides of each sample (Figure 1). For consistency throughout the trials, a single operator was responsible for delineating the baseline of the droplets across all samples. Statistical analysis was performed using a one-way analysis with variance (ANOVA) test with a significant mean difference at the 0.05 level. 

## 3. Results

Table 1 provides the mean, standard deviation, and 95% confidence interval for contact angle values on each implant disk surface among the different bovine-grafting materials solutions. Multiple comparisons of each wetting solution were performed on CP-Ti, niobium, TiZrO_2_, and zirconia, respectively (Table 2, Table 3, Table 4 and Table 5).

Commercially pure titanium exhibited mean wettability values of 93.68 ± 2.524 for Group A, 95.37 ± 1.917 for Group B, and 94.31 ± 2.433 for Group C. There was no significant difference when comparing the Group C control to both bovine grafting materials. However, a statistical significance (*p* < 0.05) was found when comparing bovine grafting materials to each other on commercially pure titanium, with Group A seeming to improve the hydrophilicity of titanium more than the Group B grafting material.

Niobium exhibited mean wettability values of 92.22 ± 3.929 for Group A, 92.09 ± 7.830 for Group B, and 90.61 ± 9.270 for Group C. No statistical significance was found (*p* > 0.05), as this surface remained consistently hydrophobic for each wetting solution.

The zirconia surface (SDS^®^) exhibited mean wettability values of 90.64 ± 4.535 for Group A, 92.78 ± 3.957 for Group B, and 76.06 ± 8.544 for Group C. The TiZrO_2_ surface presented mean wettability values of 95.83 ± 1.511 for Group A, 95.40 ± 2.480 for Group B, and 88.70 ± 4.077 for Group C. For zirconia and TiZrO_2_ disk surfaces, a statistically significant difference (*p* < 0.05) was found when comparing the Group C control to the bovine grafting materials, as Group A and Group B decreased the hydrophilicity of both TiZrO_2_ and zirconia. When comparing the two bovine grafting materials to each other, there was no significant difference between their impact on decreasing the wettability of TiZrO_2_ and zirconia.

Figure 2 presents a visual comparison of the mean contact angle values listed above for each implant disk surface. Figure 3, Figure 4, Figure 5 and Figure 6 provide a sample analysis of the average contact angle of Group A, Group B, and Group C using the Ossila^®^ goniometer on CP-Ti, niobium, TiZrO_2_, and zirconia, respectively.

**Figure 2 materials-17-04052-f002:**
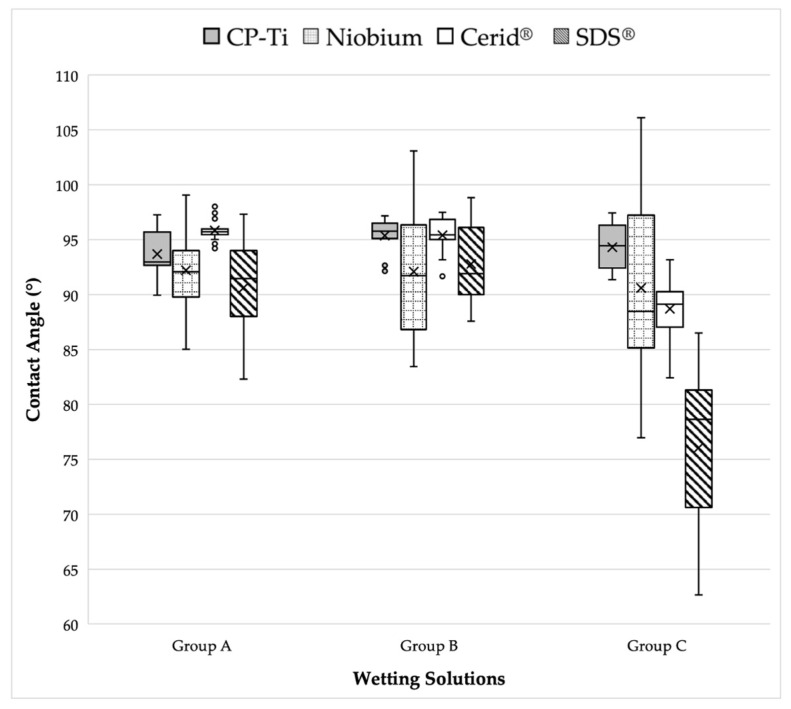
Visual comparison of mean contact angle values on CP-Ti, niobium, TiZrO_2_, and zirconia disk surfaces among different bovine grafting materials solutions.

**Table 2 materials-17-04052-t002:** Multiple comparisons of wetting solutions on CP-Ti with a mean significant difference at the 0.05 level (*). No significant difference could be shown when comparing the Group C control to Group A and Group B. However, a significant difference (*p* < 0.05) was shown when comparing the bovine grafting materials to each other with Group A seeming to improve the hydrophilicity more than Group B.

(I) Group	(J) Group	Mean Difference (I–J)	Std. Error	Sig.	95% CI
Group A	Group C	−0.63	0.596	0.887	(−2.08, 0.83)
	Group B	−1.69 *	0.596	0.017	(−3.14, −2.36)
Group B	Group C	1.06	0.596	0.233	(−0.39, 2.52)
	Group A	1.69 *	0.596	0.017	(0.24, 3.14)
Group C	Group B	−1.06	0.596	0.233	(−2.52, 0.39)
	Group A	0.63	0.596	0.887	(−0.83, 2.08)

**Table 3 materials-17-04052-t003:** Multiple comparisons of wetting solutions on niobium with a mean significant difference at the 0.05 level. No significant difference could be shown (*p* > 0.05) for all comparisons.

(I) Group	(J) Group	Mean Difference (I–J)	Std. Error	Sig.	95% CI
Group A	Group C	1.61	1.901	1.000	(−3.03, 6.25)
	Group B	0.130	1.901	1.000	(−4.51, 4.77)
Group B	Group C	1.48	1.901	1.000	(−3.16, 6.12)
	Group A	−0.13	1.901	1.000	(−4.77, 4.51)
Group C	Group B	−1.48	1.901	1.000	(−6.12, 3.16)
	Group A	−1.61	1.901	1.000	(−6.25, 3.03)

**Table 4 materials-17-04052-t004:** Multiple comparisons of wetting solutions on TiZrO_2_ with a mean significant difference at the 0.05 level (*). A significant difference could be shown (*p* < 0.05) when comparing the Group C control to Group A and Group B. No significant difference was shown when comparing the bovine grafting materials to each other.

(I) Group	(J) Group	Mean Difference (I–J)	Std. Error	Sig.	95% CI
Group A	Group C	7.14 *	0.746	<0.001	(5.32, 8.96)
	Group B	0.43	0.746	1.000	(−1.39, 2.26)
Group B	Group C	6.70 *	0.746	<0.001	(4.88, 8.52)
	Group A	−0.43	0.746	1.000	(−2.26, 1.39)
Group C	Group B	−6.70 *	0.746	<0.001	(−8.52, −4.88)
	Group A	−7.14 *	0.746	<0.001	(−8.96, −5.32)

**Table 5 materials-17-04052-t005:** Multiple comparisons of wetting solutions on zirconia with a mean significant difference at the 0.05 level (*). Similar to TiZrO_2_, a significant difference could be shown (*p* < 0.05) when comparing the Group C control to Group A and Group. No significant difference was shown when comparing the bovine grafting materials to each other.

(I) Group	(J) Group	Mean Difference (I–J)	Std. Error	Sig.	95% CI
Group A	Group C	14.57500 *	1.55794	0.000	(10.7718, 18.3782)
	Group B	−2.14567	1.55794	0.516	(−5.9488, 1.6575)
Group B	Group C	16.72067 *	1.55794	0.000	(12.9175, 20.5238)
	Group A	2.14567	1.55794	0.516	(−1.6575, 5.9488)
Group C	Group B	−16.72067 *	1.55794	0.000	(−20.5238, −12.9175)
	Group A	−14.57500 *	1.55794	0.000	(−18.3782, −10.7718)

**Figure 3 materials-17-04052-f003:**
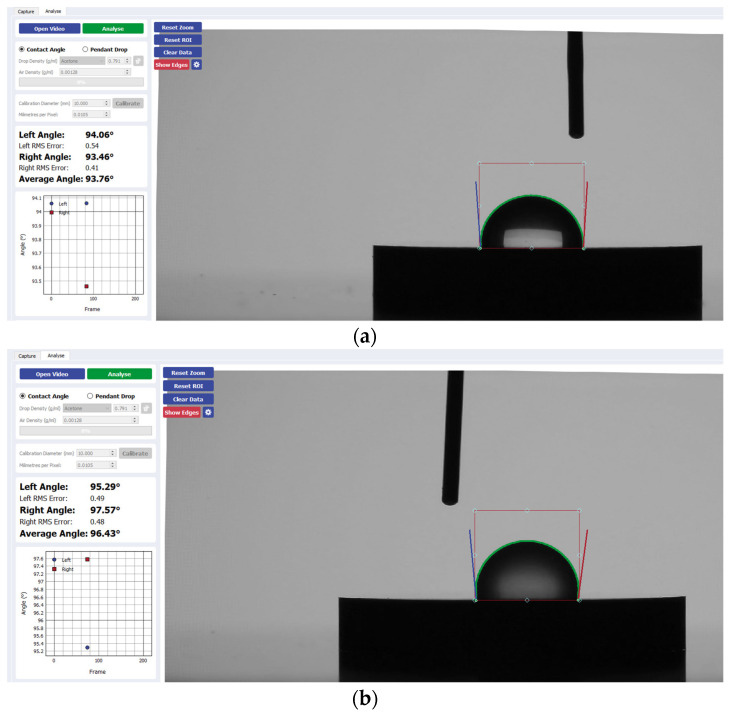
Sample analysis of the average contact angle of (**a**) Group A, (**b**) Group B, and (**c**) Group C on CP-Ti using the Ossila^®^ goniometer (Ossila Ltd, Sheffield, UK).

**Figure 4 materials-17-04052-f004:**
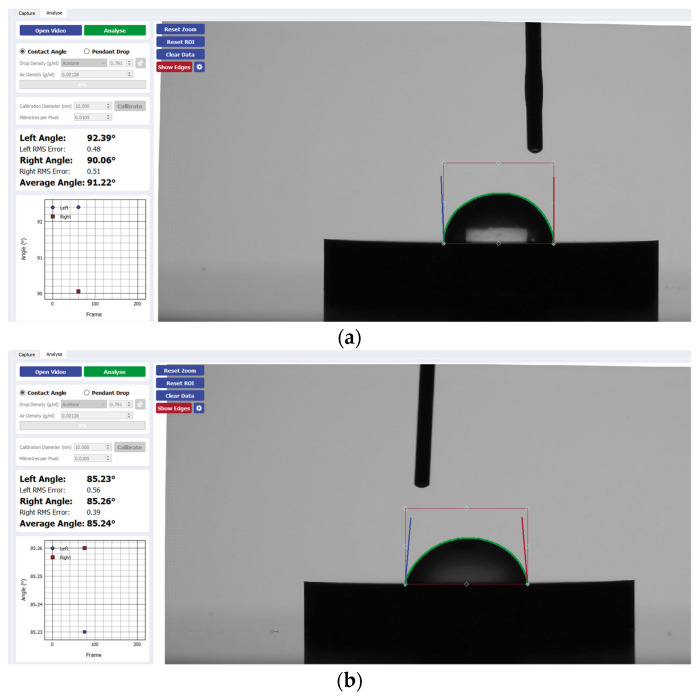
Sample analysis of the average contact angle of (**a**) Group A, (**b**) Group B, and (**c**) Group C on niobium using the Ossila^®^ goniometer.

**Figure 5 materials-17-04052-f005:**
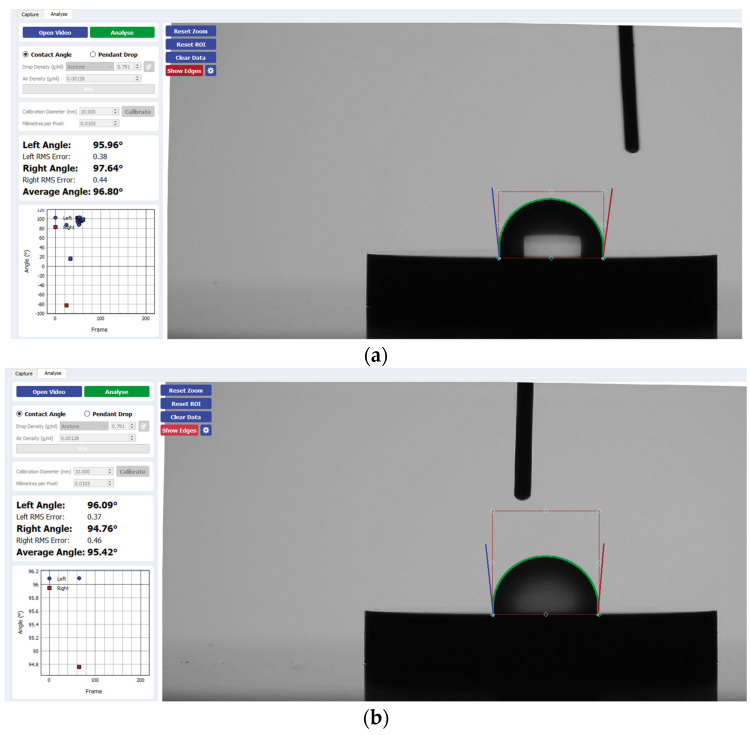
Sample analysis of the average contact angle of (**a**) Group A, (**b**) Group B, and (**c**) Group C on TiZrO_2_ using the Ossila^®^ goniometer.

**Figure 6 materials-17-04052-f006:**
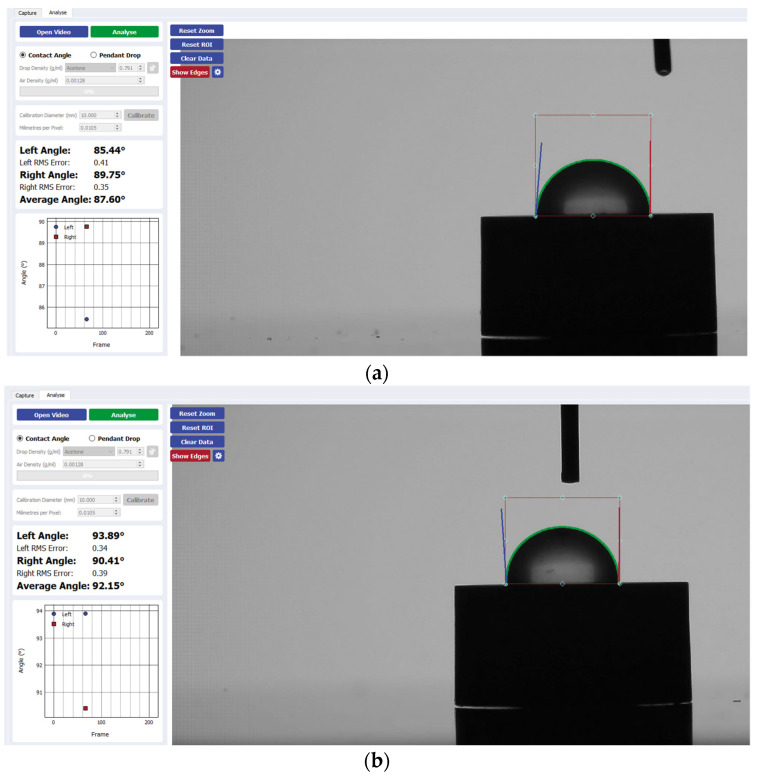
Sample analysis of the average contact angle of (**a**) Group A, (**b**) Group B, and (**c**) Group C on zirconia using the Ossila^®^ goniometer.

## 4. Discussion

The present study proposed to test the null hypothesis that there is no difference in the hydrophilicity of implant disk surfaces when comparing bovine grafting materials, after hydration, to the saline control. The null hypothesis is rejected for TiZrO_2_ and zirconia, as this study found a statistically significant decrease in hydrophilicity when these surfaces were treated with bovine grafting materials (Group A and B) compared to the control (Group C). For pure titanium, the grafting materials had no significant effect on wettability compared to the control. However, when comparing the xenografts to each other, Group A seemed to improve the hydrophilicity more than Group B. We fail to reject the null hypothesis for niobium, as no statistically significant difference in wettability was observed when comparing the grafting materials to the control, and all groups were consistently hydrophobic.

Based on previous in vitro studies from our laboratory, we can confirm that certain bovine grafting materials might have a negative impact on cell proliferation due to toxicological characteristics. In a previous study, when bone grafting materials including Bio-Oss^®^, mineralized cancellous human bone (AMBA), and natural hydroxyapatite (HA) were cultured on tissue culture plastics, they seemed to exert a toxic effect causing the cell death of human primary osteosarcoma osteoblast-like SaOS_2_ cells. This occurred at 0.025 g/mL, resulting in a reduction of at least 50% of the cells. The number of cells took 48 h to reduce to half of the originally plated number [18]. A more recent study analyzed bone grafts from synthetic, bovine, and human sources to test for cytotoxicity on dental pulp stem cells (DPSCs) and osteosarcoma cells. This study revealed large concentrations of tungsten (W) via X-ray fluorescence as high as 400 ppm in synthetic bovine grafting materials that was proven to be highly cytotoxic when directly exposed to both human DPSC and SaOS_2_ cells [19]. The fact that harmful concentrations of tungsten are present in commercially available bovine grafting materials warrants concern for their potential in vivo systemic toxicological impact during the treatment of peri-implantitis defects. This may contribute to the unpredictable clinical outcomes in peri-implantitis treatment, as these products are frequently used in routine clinical practice during implant procedures.

This present study looks at the current standard of care which includes mixing commercially available bovine grafting materials in a saline solution. Based on the results from the present in vitro study, certain bovine grafting materials after mixing with saline solutions may negatively influence the hydrophilicity of zirconia surfaces. This may make some grafting materials and solutions less favorable for the treatment of peri-implantitis intrabony defects. This recognition of hydrophobic properties introduced by xenografts has a long-term benefit for clinical research, as these findings may warrant the development of new treatment protocols to mitigate these effects. This could include the generation of novel pre-treatment modalities for implants to counteract the hydrophobic properties induced by bovine grafting or the use of alternative grafting materials. Additionally, the findings from this study can help clinicians enhance their criteria for selecting grafting materials, especially in patients with a higher risk of developing peri-implantitis.

Certainly, there is a requirement for further in vivo studies in order to make this statement clinically relevant. Future studies should also investigate wetting solutions other than saline to confirm if the unfavorable effects are derived from the bovine grafting materials themselves rather than the solution in which they are suspended. Other solutions in future studies may include bovine serum albumin (BSA) and/or blood products.

The bovine grafting materials in this study did not seem to affect the hydrophilicity of the metal implant surfaces compared to the controls. However, when comparing the grafting materials to each other, there are differences in the hydrophilicity of implant surfaces after hydration of the grafting materials. Therefore, the material properties and the biological effects of the bone grafting materials should be extensively studied when peri-implantitis defects around titanium dental implants have to be treated.

It is widely accepted that surface wettability plays a fundamental role in bacterial adhesion. Therefore, researchers attempting to improve the functional conditions of implants should assess wettability as a parameter to determine the anti-bacterial behavior of an implant surface and ultimately prevent the development of peri-implantitis. This is because multiple previous studies have shown a relationship between microbial plaque colonization and the pathogenic development of peri-implant infections [20,21]. Additionally, biomaterials with adequate hydrophilicity have an improved biocompatibility and increased cell growth, thereby enhancing interactions between implant surfaces and the biological environment. Surface wettability can affect the long-term stability of implants and therefore should be further studied to better understand the interface between bone, implant, and biomaterial surfaces [22].

Oftentimes, the successful osseointegration of dental implants can be undermined by alveolar bony defects. This includes alveolar bone dehiscences and fenestrations related to the insertion of implants, which can be caused by inadequate buccolingual bone width and the inaccurate positioning of implants during placement [23]. Therefore, bovine-derived substitutes are the most commonly used xenografts for dental implant surgery to compensate for bone loss and enhance the edentulous ridge [24]. However, there is minimal existing research on the effects of bovine grafting materials on the hydrophilic behavior of both traditional and novel dental implant surface materials. While this study aims to provide more insight into the interactions of two xenografts after hydration with saline, and their impact on the hydrophilicity of dental implants, future research is necessary for a more comprehensive understanding of their interactions and clinical consequences.

The surface energy changes and the interaction between the bone grafting biomaterial and the implant surface are basic principles in order to have successful augmentative procedures in peri-implant defects especially in the treatment of peri-implantitis. More studies in this field are required to evaluate the hydrophilicity of implant surfaces and how this can be increased using bone grafting techniques in order to improve clinical outcomes and provide reliable treatment methods.

A limitation of this study is that it only focuses on wettability testing of the implant surfaces. In conjunction with wettability, further experiments can also incorporate a correlation analysis with other surface properties including surface roughness analysis, scanning electron microscopy (SEM) investigations, and X-ray photoelectron spectroscopy (XPS) surface chemistry analysis. This will enable a more comprehensive understanding of how implant surfaces may interact with the biological environment. Additionally, in regard to zirconia-coated implant surfaces like TiZrO_2_ (Cerid^®^), the long-term sustainability of these coatings should be further investigated for their clinical application.

## 5. Conclusions

The aim of the present study was to examine the effect of two bovine grafting materials, after hydration, on the wettability of implant disk surfaces manufactured of commercially pure titanium, TiZrO_2_, zirconia, and niobium in order to investigate the reasons behind the inconsistent reliability of peri-implantitis treatment. Within the limitations of the present in vitro study, it can be concluded that bovine grafting materials after hydration may decrease the hydrophilicity of zirconia surfaces and have hydrophobic properties on titanium surfaces. All this can have a negative impact on the predictable outcomes of peri-implantitis therapy. Therefore, future investigation is necessary to determine how hydrophobic properties affect the long-term stability and performance of dental implants. This may include the further evaluation of bovine grafting adhesion to implant surfaces over extended periods of time under various physiological conditions.

## Figures and Tables

**Figure 1 materials-17-04052-f001:**
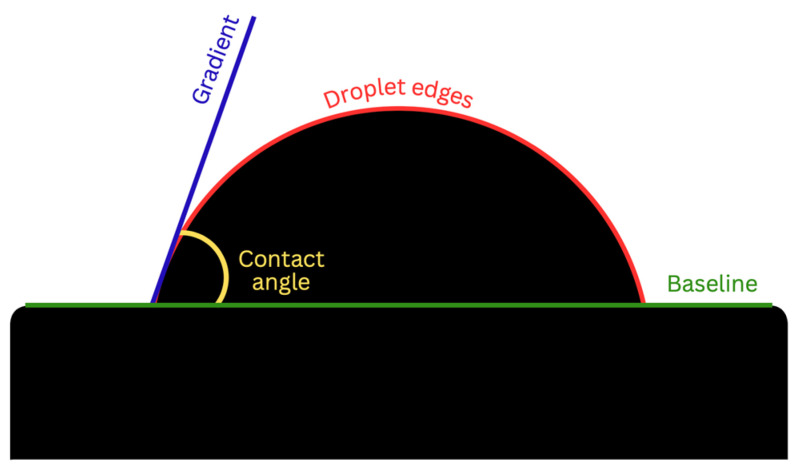
Contact angle analysis via the sessile drop technique.

**Table 1 materials-17-04052-t001:** Mean, standard deviation, and 95% confidence interval for contact angle values on CP-Ti, niobium, TiZrO_2,_ and zirconia disk surfaces among different bovine grafting materials solutions (after graft hydration).

Wetting Solution and Surface		N	Mean	Std. Deviation	95% CI
Group A	CP-Ti	30	93.68	2.524	(92.74, 94.62)
	Niobium	30	92.22	3.929	(90.75, 93.68)
	Cerid^®^	30	95.83	1.511	(95.27, 96.40)
	SDS^®^	30	90.64	4.535	(88.95, 92.33)
Group B	CP-Ti	30	95.37	1.917	(94.66, 96.09)
	Niobium	30	92.09	7.830	(89.16, 95.01)
	Cerid^®^	30	95.40	2.480	(94.47, 96.32)
	SDS^®^	30	92.78	3.957	(91.31, 94.26)
Group C	CP-Ti	30	94.31	2.433	(93.40, 95.22)
	Niobium	30	90.61	9.270	(87.15, 94.07)
	Cerid^®^	30	88.70	4.077	(87.17, 90.22)
	SDS^®^	30	76.06	8.544	(72.87, 79.25)

## Data Availability

Data are contained within the article.

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
