# Peer review of "Bovine Mineral Grafting Affects the Hydrophilicity of Dental Implant Surfaces: An In Vitro Study"

_materials, 2024, doi:10.3390/ma17164052_

Round 1

Reviewer 1 Report

Comments and Suggestions for Authors

This study investigated how different bone grafting materials influence the wettability of various implant surface materials. While the study is well-structured, its originality is limited.

Introduction

The introduction provides a comprehensive overview of dental implant materials, surface properties, and the role of bone grafting materials. It also covers recent advancements in the field, including novel materials like zirconia and titanium-zirconium dioxide composites. However, this section is overly extensive and could be condensed.

Methods

The methodology is well-defined and adequately detailed.

Results

A slight increase in hydrophilicity was observed for Group A on titanium, but this difference was not statistically significant. This interesting finding could be attributed to factors such as high data variability or a small sample size. To clarify this result, further investigation with larger sample sizes.

Discussion

While the discussion adequately describes the results, it falls short in providing a deeper understanding of the underlying mechanisms responsible for the observed changes in wettability. A correlation analysis between surface roughness, chemical composition, and wettability could offer valuable insights into these mechanisms.

Author Response

Thank you very much for taking the time to review this manuscript. Please find the detailed responses below and the corresponding revisions highlighted in the re-submitted file.

Comment 1 (Introduction): The introduction provides a comprehensive overview of dental implant materials, surface properties, and the role of bone grafting materials. It also covers recent advancements in the field, including novel materials like zirconia and titanium-zirconium dioxide composites. However, this section is overly extensive and could be condensed.

Response 1: Thank you for pointing this out. We agree with this comment. Therefore, we have condensed this section by removing information we no longer deemed necessary for the introduction (information from lines 44-45, 64-73, 80-81, and 92-100 has been omitted).

Comment 2 (Results): A slight increase in hydrophilicity was observed for Group A on titanium, but this difference was not statistically significant. This interesting finding could be attributed to factors such as high data variability or a small sample size. To clarify this result, further investigation with larger sample sizes.

Response 2: The increase in hydrophilicity observed for Group A on titanium was found to be statistically significant. This is stated in lines 188-191: “However, a statistical significance (p < 0.05) was found when comparing bovine grafting materials to each other on commercially pure titanium, with Group A seeming to improve the hydrophilicity of titanium more than Group B grafting material.”

Comment 3 (Discussion): While the discussion adequately describes the results, it falls short in providing a deeper understanding of the underlying mechanisms responsible for the observed changes in wettability. A correlation analysis between surface roughness, chemical composition, and wettability could offer valuable insights into these mechanisms.

Response 3: Agree. We have, accordingly, revised the last paragraph of the discussion to emphasize this point: “A limitation of this study is that it only focuses on wettability testing of the implant surfaces. In conjunction with wettability, further experiments can also incorporate a correlation analysis with other surface properties including surface roughness analysis, scanning electron microscopy (SEM) investigations, and X-ray photoelectron spectroscopy (XPS) surface chemistry analysis. This will enable a more comprehensive understanding of how implant surfaces may interact with the biological environment” (lines 373-379).

Reviewer 2 Report

Comments and Suggestions for Authors

In the paper titled “Bovine Mineral Grafting Affects the Hydrophilicity of Dental Implant Surfaces: An In Vitro Study” investigated the influence of two bovine grafting materials after hydration on the wettability different disk surfaces: commercially pure titanium, titanium-zirconium dioxide, zirconia and niobium.

Abstract:

-  No statistical analysis was informed.

- No sample size was informed.

- I would recommend refining the content of the abstract and providing a general summary of the results rather than writing the limitations of the study.

Introduction:

- What are the hypotheses to be tested in the research?

Materials and Methods:

- It is relevant that the proposed methodology be based on previous studies or scientific papers in the literature.

- How did the authors base themselves on obtaining the sample number for experimental tests? It would be beneficial to go into more details.

- I suggest that the authors rename the groups (A, B and C), to allow for easier understanding by the reader.

Results

- Figures 2, 3, 4, 5, 6 and 7 are not cited in the text, I recommend that the authors rewrite the results section to include them or remove them from the manuscript.

Discussion

- In the first two paragraphs of the discussion, a general and detailed analysis of the findings must be carried out, in addition to reporting whether the null hypotheses were accepted or rejected.

- Although this is a study that portrays a new material for bone grafting, the discussion is merely informative. It is necessary for the authors to discuss their results in more depth. Therefore, I suggest that the authors shall compare similar results in literature.

- What are the authors' future perspectives on this topic? How can their findings benefit clinical research in the long term?

- What are the limitations of the study? Discussing methodological limitations that may affect the

interpretation of results would add depth to the study.

Conclusion

- Limitations should be addressed in the discussion section.

- Rewrite the conclusion, highlighting the main results obtained in the study and the authors interpretation.

- Caution is required with statements obtained from the results of an in vitro study

Throughout the manuscript, some sentences were not referenced by the authors, making it necessary to reference them.

Author Response

Thank you very much for taking the time to review this manuscript. Please find the detailed responses below and the corresponding revisions highlighted in the re-submitted files.

Comment 1 (Abstract): No statistical analysis was informed. No sample size was informed. I would recommend refining the content of the abstract and providing a general summary of the results rather than writing the limitations of the study.

Response 1: Thank you for pointing this out. We agree with this comment. Therefore, we have revised this section to include information about statistical analysis and sample size (lines 18-20). Additionally, a general summary of the results are provided without discussing limitations of the study (lines 24-29).

Comment 2 (Introduction): What are the hypotheses to be tested in the research?

Response 2: We have revised the introduction to include information at the end of this section that explains our hypothesis testing: “Therefore, the aim of this in-vitro study was to test the null hypothesis that bovine grafting materials, after hydration, have no effect on the hydrophilicity of CP-Ti, TiZrO2, zirconia, and niobium implant disk surfaces compared to the saline control” (lines 116-119).

Comment 3 (Methods): It is relevant that the proposed methodology be based on previous studies or scientific papers in the literature. How did the authors base themselves on obtaining the sample number for experimental tests? It would be beneficial to go into more details.

Response 3: We have revised the materials and methods section to better explain our sample number using prior power analysis: “This resulted in 12 experimental groups, each composed of 15 repetitions (N = 360). Contact angle measurements were obtained on the left and right sides of each droplet. An a prior power analysis was conducted using Cohen (1992) to determine the minimum sample size required to test the study hypothesis. Results indicated the required sample size to achieve 80% power for detecting a medium/large effect, at a significance criterion of α = .05, was N = 30 for ANOVA [17]” (lines 159-164).

Comment 4 (Methods): I suggest that the authors rename the groups (A, B and C), to allow for easier understanding by the reader.

Response 4: Thank you for this suggestion. We used groups A, B and C in order to avoid the mentioning of brand names throughout the study.

Comment 5 (Results): - Figures 2, 3, 4, 5, 6 and 7 are not cited in the text, I recommend that the authors rewrite the results section to include them or remove them from the manuscript.

Response 5: Agree. We have, accordingly, modified the text to include to include these figures in the manuscript (lines 203-206).

Comment 6 (Discussion): In the first two paragraphs of the discussion, a general and detailed analysis of the findings must be carried out, in addition to reporting whether the null hypotheses were accepted or rejected.

Response 6: Agree. We have, accordingly, modified this section to include a paragraph discussing the results of the study: “The present study proposed to test the null hypothesis that there is no difference in the hydrophilicity of implant disk surfaces when comparing bovine grafting materials, after hydration, to the saline control. The null hypothesis is rejected for TiZrO2 and zirconia, as this study found a statistically significant decrease in hydrophilicity when these surfaces were treated with bovine grafting materials (Group A and B) compared to the control (Group C). For pure titanium, the grafting materials had no significant effect on wettability compared to the control. However, when comparing the xenografts to each other, Group A seemed to improve the hydrophilicity more than Group B. We fail to reject the null hypothesis for niobium, as no statistically significant difference in wettability was observed when comparing the grafting materials to the control, and all groups were consistently hydrophobic” (lines 294-304).

Comment 7 (Discussion): Although this is a study that portrays a new material for bone grafting, the discussion is merely informative. It is necessary for the authors to discuss their results in more depth. Therefore, I suggest that the authors shall compare similar results in literature.

Response 7: Thank you for pointing this out. We understand the importance of comparing prior results in the literature to provide context and depth to our discussion. This is why we have discussed results in the literature from Huda (2018) and Yang et al. (2022) in this section (lines 305-321). After a thorough review of the existing literature, we have found no previous studies that have tested our surfaces with similar bone grafting wetting solutions in our experiment. Therefore, in other parts of this section we have focused more on the implications of our findings.

Comment 8 (Discussion): What are the authors' future perspectives on this topic? How can their findings benefit clinical research in the long term?

Response 8: We have revised the discussion section to address these questions: “Based on the results from the present in-vitro study, certain bovine grafting materials after mixing with saline solutions may influence negatively the hydrophilicity of zirconia surfaces. This may make some grafting materials and solutions less favorable for the treatment of peri-implantitis intrabony defects. This recognition of hydrophobic properties introduced by xenografts has a long-term benefit for clinical research, as these findings may warrant the development of new treatment protocols to mitigate these effects. This could include the generation of novel pre-treatment modalities for implants to counteract the hydrophobic properties induced by bovine grafting, or the use of alternative grafting materials. Additionally, the findings from this study can help clinicians to enhance their criteria for selecting grafting materials, especially in patients with a higher risk of developing peri-implantitis” (lines 323-333).

Comment 9 (Discussion): What are the limitations of the study? Discussing methodological limitations that may affect the interpretation of results would add depth to the study.

Response 9: Agree. We have, accordingly, revised the last paragraph of the discussion to emphasize this point: “A limitation of this study is that it only focuses on wettability testing of the implant surfaces. In conjunction with wettability, further experiments can also incorporate a correlation analysis with other surface properties including surface roughness analysis, scanning electron microscopy (SEM) investigations, and X-ray photoelectron spectroscopy (XPS) surface chemistry analysis. This will enable a more comprehensive understanding of how implant surfaces may interact with the biological environment”(lines 373-379).

Comment 10 (Conclusion): Rewrite the conclusion, highlighting the main results obtained in the study and the authors interpretation.

Response 10: Agree. We have, accordingly, revised the conclusion section to highlight the main results and offer our interpretation that this may have a negative impact the predictability of peri-implantitis therapy. The conclusion includes the following: “The aim of the present study was to examine the effect of two bovine grafting materials, after hydration, on the wettability of implant disk surfaces manufactured of commercially pure titanium, TiZrO2, zirconia, and niobium. This was done in order to investigate the reasons behind the inconsistent reliability of peri-implantitis treatment. Within the limitations of the present in-vitro study, it can be concluded that bovine grafting materials after hydration may decrease the hydrophilicity of zirconia surfaces and have hydrophobic properties on titanium surfaces. All this can have negative impact on the predictable outcomes of peri-implantitis therapy. Therefore, future investigation is necessary to determine how hydrophobic properties affect the long-term stability and performance of dental implants. This may include further evaluation of bovine grafting adhesion to implant surfaces over extended periods of time under various physiological conditions” (lines 383-394).

Comment 11 (Conclusion): Caution is required with statements obtained from the results of an in vitro study.

Response 11: Agree. We have emphasized this in the conclusion by stating the following: “Within the limitations of the present in-vitro study, it can be concluded that bovine grafting materials after hydration may decrease the hydrophilicity of zirconia surfaces and have hydrophobic properties on titanium surfaces” (lines 386-389).